# Investigation of hs-TnI and sST-2 as Potential Predictors of Long-Term Cardiovascular Risk in Patients with Survived Hospitalization for COVID-19 Pneumonia

**DOI:** 10.3390/biomedicines10112889

**Published:** 2022-11-10

**Authors:** Lukas Fiedler, Lukas J. Motloch, Peter Jirak, Ruslan Gumerov, Paruir Davtyan, Diana Gareeva, Irina Lakman, Alexandr Tataurov, Gulnaz Lasinova, Valentin Pavlov, Laurenz Hauptmann, Kristen Kopp, Uta C. Hoppe, Michael Lichtenauer, Rudin Pistulli, Anna-Maria Dieplinger, Naufal Zagidullin

**Affiliations:** 1University Clinic for Internal Medicine II, Paracelsus Medical University, 5020 Salzburg, Austria; 2Department of Internal Medicine, Cardiology, Nephrology and Intensive Care Medicine, Hospital Wiener Neustadt, 2700 Wiener Neustadt, Austria; 3Department of Internal Diseases, Bashkir State Medical University, Lenin Str. 3, 450008 Ufa, Russia; 4Scientific Laboratory for the Socio-Economic Region Problems Investigation, Ufa University of Science and Technology, Zaki Validi Str. 32, 450076 Ufa, Russia; 5Department of Biomedical Engineering, Ufa University of Science and Technology, Zaki Validi Str. 32, 450076 Ufa, Russia; 6Department of Urology, Bashkir State Medical University, Lenin Str. 3, 450008 Ufa, Russia; 7Department of Cardiology I, Coronary and Peripheral Vascular Disease, Heart Failure, University Hospital Muenster, 48149 Muenster, Germany; 8Nursing Science Program, Institute for Nursing Science and Practice, Paracelsus Medical University, 5020 Salzburg, Austria; 9Medical Faculty, Johannes Kepler University Linz, 4040 Linz, Austria

**Keywords:** MACE 1, cardiovascular death 2, COVID-19 3, post-acute COVID-19 sequelae 4, long COVID-19 5, troponin 6, sST-2 7

## Abstract

Introduction: COVID-19 survivors reveal an increased long-term risk for cardiovascular disease. Biomarkers like troponins and sST-2 improve stratification of cardiovascular risk. Nevertheless, their prognostic value for identifying long-term cardiovascular risk after having survived COVID-19 has yet to be evaluated. Methods: In this single-center study, admission serum biomarkers of sST-2 and hs-TnI in a single cohort of 251 hospitalized COVID-19 survivors were evaluated. Concentrations were correlated with major cardiovascular events (MACE) defined as cardiovascular death and/or need for cardiovascular hospitalization during follow-up after hospital discharge [FU: 415 days (403; 422)]. Results: MACE was a frequent finding during FU with an incidence of 8.4% (cardiovascular death: 2.8% and/or need for cardiovascular hospitalization: 7.2%). Both biomarkers were reliable indicators of MACE (hs-TnI: sensitivity = 66.7% & specificity = 65.7%; sST-2: sensitivity = 33.3% & specificity = 97.4%). This was confirmed in a multivariate proportional-hazards analysis: besides age (HR = 1.047, 95% CI = 1.012–1.084, *p* = 0.009), hs-TnI (HR = 4.940, 95% CI = 1.904–12.816, *p* = 0.001) and sST-2 (HR = 10.901, 95% CI = 4.509–29.271, *p* < 0.001) were strong predictors of MACE. The predictive value of the model was further improved by combining both biomarkers with the factor age (concordance index hs-TnI + sST2 + age = 0.812). Conclusion: During long-term FU, hospitalized COVID-19 survivors, hs-TnI and sST-2 at admission, were strong predictors of MACE, indicating both proteins to be involved in post-acute sequelae of COVID-19.

## 1. Introduction

COVID-19 represents a major health burden worldwide. The pathology of acute coronavirus disease 2019 (COVID-19) entails bilateral pneumonia, acute respiratory distress syndrome, and cardiovascular events. The clinical management of COVID-19 patients represents a challenge, despite the increasing knowledge regarding disease progression in high-risk patients [1]. The mechanism of infection of severe acute respiratory syndrome coronavirus type 2 (SARS-CoV-2) has been identified, with its spike glycoprotein binding to the angiotensin-converting enzyme 2 (ACE-2) receptor of host cells [2]. Therefore, cells exhibiting a high expression of the ACE-2 receptor, such as type II alveolar cells, myocardial cells, and endothelial and artery smooth muscle cells, amongst others, are particularly prone to causing clinical symptoms of COVID-19 [3]. The expression of ACE-2 on cells involved in local and systemic hemodynamics may explain the impact of SARS-CoV-2 infection on the cardiovascular system [4]. Of note, cardiovascular symptoms including not only cardiac injury, but also arrhythmic and thromboembolic events, are established disease-specific manifestations of acute COVID-19 disease [5,6,7,8].

In the early phase of the pandemic, acute COVID-19 disease lead to overstrained medical systems, affecting millions of individuals worldwide and long-term sequelae of COVID-19 represents a major health burden around the globe [9]. Following the acute phase of disease, COVID-19 survivors may experience persistent symptoms for a longer time period affecting different organ systems, classified as long-term sequelae of COVID-19 [10]. With millions of infected worldwide, this syndrome binds major medical resources around the globe. In this context, recent data also revealed an increased risk for not only thromboembolic events but also for various cardiovascular pathologies, including arrhythmias, myocardial infarction, myocarditis, thromboembolic events, and heart failure in COVID-19 survivors even during long-term follow-up. Consequently, COVID-19 survivors seem to be at increased risk for cardiovascular events emphasizing the need for novel cardiovascular risk assessment strategies to improve medical care in this population. However, while cardiovascular risk during long-term follow-up correlates with severity of acute COVID-19 disease [4], specific risk assessment strategies for COVID-19 survivors are missing.

Despite the established correlation between myocardial injury and mortality [11], prognostic cardiovascular biomarkers to predict the cardiovascular health of COVID-19 patients are scarce. Serum cardiac high-sensitive troponin I (hs-TnI) is the gold standard marker for cardiovascular risk assessment [12]. Increased levels of this protein have been positively correlated with mortality and severe disease course in COVID-19 patients [13,14,15]

A potential novel cardiovascular marker is suppression of tumorigenicity-2 (ST-2), a member of the interleukin-1 receptor family [16,17,18]. Two isoforms of this protein have been described: a membrane-bound ligand, ST2L, and a soluble form, sST-2 [19]. Both isoforms have been implicated in COVID-19 pathogenesis [20,21,22]. In this context, high sST-2 levels appear to correlate with serum levels of C-reactive protein (CRP), a classical marker of inflammation, with severe disease course and requirement for intensive care of COVID-19 patients, as well as with COVID-19-associated mortality [23,24].

Although the involvement of sST-2 and hs-TnI in COVID-19 pathogenesis and inflammatory progresses has been documented, their potential as prognostic markers of cardiovascular events, especially during long-term follow-up, has not been comprehensively evaluated to date in COVID-19 survivors.

Therefore, the aim of the present study was to assess sST-2 and hs-TnI levels of hospitalized COVID-19 patients with pneumonia and, in the same cohort, correlate their serum levels with major cardiovascular events (MACE) after hospital discharge during long-term follow-up. We hypothesized that in COVID-19, sST-2, and hs-TnI are suitable cardiovascular biomarkers to predict long-term cardiovascular outcomes, indicating a promising risk assessment strategy in the hospitalized COVID-19 population.

## 2. Materials and Methods

### 2.1. Study Design

The present study was performed in accordance with standards of good clinical practice and the principles of the Declaration of Helsinki. The study protocol was approved by the Local Ethical Committee (N5, 2020), and written informed consent was obtained from all participants before study inclusion.

In this prospective, non-randomized trial, we screened a single cohort of 257 consecutive patients presenting with moderate COVID-19 pneumonia not requiring mechanical ventilation in a single tertiary center between June 2020 and September 2020. Diagnosis of COVID-19 disease was verified upon admission using PCR testing, specific antibodies, and CT-scan.

All included patients were 18 years or older, presenting with moderate COVID-19 pneumonia. To account for disease severity and associated potential cardiovascular risk, patients who developed severe COVID-19 pneumonia requiring mechanical ventilation during their hospitalization were excluded from further analyses. Further exclusion criteria included acute ST-elevation myocardial infarction at admission, acute stroke at admission, active malignant disease within the last three years, and acute kidney failure at admission defined as glomerular filtration rate (GFR) < 30 mL/min/1.73 m^2^, as well as pregnancy or lactation. Follow-up and further analyses were performed in hospital survivors only. Based on these exclusion criteria, six patients were excluded from further analyses.

Patient enrollment and the design of the study are presented in Figure 1. Venous blood was drawn upon hospital admission and subsequently centrifuged. The remaining serum was frozen at −20 °C for further analyses. Analyses of sST-2 and hs-TnI concentrations were performed as indicated by the manufacturer using enzyme immunoassays (Critical diagnostics, San Diego, CA, USA, for sST-2 and Hema Ltd., Tower Mumbai, Russia, for hs-TnI).

COVID-19 diagnostics and specific medical treatment were administered according to current national COVID-19 guidelines [25]. Upon admission, a detailed medical history (current symptoms, previous illnesses, and current medications) was obtained from all study participants and clinically relevant laboratory parameters were routinely measured (Table 1). Follow-up for the study endpoints was conducted after hospital discharge for a median of 415 days (403; 422). Patients were followed up using the regional medical information analytical system “ProMed” [26]. This is a web-based medical records system which ensures remote online monitoring of hospitalization discharge notes, as well as death certificates. Follow-up endpoints were analyzed in October 2021.

The study endpoint MACE was defined as cardiovascular death and/or the need for cardiovascular hospitalization, including hospitalization due to pulmonary embolism, stroke, cardiac arrhythmias, myocardial infarction, acute coronary syndrome, and/or heart failure. Cardiovascular death was defined as in-hospital death due to cardiovascular causes or out of hospital death meeting the criteria of sudden cardiac death [27].

### 2.2. Statistical Analyses

The mathematical model of our statistical analyses is summarized in Figure 2. Primarily all data were tested for normal distribution using the Shapiro–Wilk test. If normally distributed, continuous variables were presented as mean values (M) and standard deviations (SD), or if non-normally distributed, data were expressed as medians (interquartile range Q1–Q3). For non-parametric data, the Mann–Whitney test was applied for comparison between two groups. Categorical variables were presented as frequencies. Proportions and differences among groups were tested using the Chi-squared test. If the frequency of a variable in one of the groups was low or absent, the Chi-squared criterion was adjusted for likelihood.

To evaluate the predictive power for cardiovascular events in COVID-19 survivors, univariate Cox proportional hazards models were calculated at the preliminary stage of the analysis. The dependent variable in such regressions was set as the time until MACE and the independent variable as the studied covariate. Since cardiovascular risk was explored for a longer follow-up over a time period of one year, adjustments were made for age. The predictor of MACE was considered statistically significant if the null hypothesis was equal to zero, with the corresponding coefficient in the Cox regression equation rejected at a value of *p* < 0.05.

The investigated biomarkers variables sST-2 and hs-TnI have been preliminarily binarized: the binarized biomarker variable was equal to 0 if the biomarker value did not exceed the cut-off threshold, and equal to 1 if the value was above the cut-off. The cut-off points were determined according to ROC analysis for sST-2 and for hs-TnI (see below). To assess the differences in the risk for MACE for sST-2 and hs-TnI in groups according to the upper/lower than cut-off threshold, Kaplan–Meier curves were constructed with a confidence interval according to the Greenwood formula (Figure 1 and Figure 2). The analysis of regression modeling results was carried out based on the calculation of odds ratios *HR* (*x_i_*) for each statistically significant *i*-predictor *x_i_*:$$HR\left(x_{i}\right)=\frac{h(t|x_{i})}{h_{0}\left(t\right)}=\exp\left(x_{i}\beta \right),$$

Defined as —the risk in patients with *x_i_* risk factors for incidence of MACE, *h*_0_ (*t*)—the basic (unconditional) risk for MACE during total follow-up and —Cox regression coefficient estimated by the partial likelihood in Breslow estimates.

The predictive power of the investigated model was confirmed by the Harrell concordance CI index, while R^2^mer values of the explained risk and R^2^mev values of the explained variation were applied as model quality metrics. The interpretation of the model’s results was based on the hazard ratio (*HR*) of MACE events for each risk predictor. To reinsure the predictive value of the studied serum cardiovascular biomarker as reliable predictors of MACE, three variants of models were considered: 1. A model which included all significant clinical and laboratory risk factors combined with hs-TnI, 2. A model which included all significant clinical and laboratory risk factors combined with sST-2, and 3. A model which included all significant clinical and laboratory risk factors combined with both biomarkers hs-TnI and sST-2. The predictive value of the investigated models was compared by quality metrics using the Harrell concordance CI index and measure of explained risk of models with the inclusion/exclusion of biomarker values.

A *p*-value < 0.05 was considered as statistically significant. All statistical analyses were assessed using R software (version 3.6.3, R Foundation for Statistical Computing, Vienna, Austria, https://www.r-project.org (accessed on 1 September 2022).

## 3. Results

Baseline characteristics, including demographics, clinical presentation at admission, relevant concomitant disease, and relevant laboratory parameters of the investigated cohort are presented in Table 1. Specific in-hospital COVID-19 therapies are listed in Appendix A while relevant cardiovascular post-discharge therapies are presented in Appendix A.

Follow-up was conducted for a median of 415 (403; 422) days after hospital discharge. During this time period, MACE was observed in 8.3% (*n* = 21) of the investigated patients. While 6.7% (*n* = 18) of the patients were hospitalized due to a cardiovascular pathology, 2.4% (*n* = 6) died during follow-up due to a cardiovascular cause (Table 2). MACE during follow-up, was more often observed in older patients with higher systolic and diastolic arterial blood pressure at admission. In addition, admission hs-TnI levels were significantly increased in patients who showed cardiovascular pathologies during long-term follow-up (Table 3).

Further statistical evaluation of our cohort was carried out using the described mathematic model. According to the incidence of MACE during follow-up, cut-off values for the investigated biomarkers were estimated by means of ROC analysis (Figure 1 and Figure 2). Analysis showed significant difference in event-free survival functions between under and upper cut-off values for both investigated biomarkers: hs-TnI (>0.04 ng/mL with Sensitivity = 66.7% and Specificity = 65.7, Figure 1) and sST-2 (>134.16 ng/mL with Sensitivity = 33.3% and Specificity = 97.4, Figure 2).

Kaplan–Meier event-free survival curves were created for the incidence of MACE occurrence during follow-up, which assessed under and over the cut-off values for the investigated biomarkers hs-TnI and sST-2. Prominent discrepancies in event-free survival between under- and over-curve frequency of MACE were observed for both investigated biomarkers (Figure 3, Figure 4, Figure 5 and Figure 6).

The endpoint MACE during long-term follow-up after hospital discharge was further analyzed by univariate Cox regression for each biomarker using age as the control variable. Of note, besides age, only both investigated biomarkers revealed a satisfactory significant prediction quality (*p* < 0.05): hs-TnI (HR = 3.385, 95% CI = 1.351–8.487, *p* = 0.009) and sST-2 (HR = 13.695, 95% CI = 5.441–34.477, *p* < 0.001; Appendix A).

Based on univariate analysis results, we selected only significant predictive factors for MACE during follow-up and constructed the multifactor model accordingly. Although the variable “male gender” was not significant, we decided to include it into the multi-marker model as a control variable to minimize the shift of results promoted by gender disproportions. The quality metrics of the model were estimated using Harrel’s index of concordation (CI), measure of explained randomness (R^2^mer), and measure of explained variation (R^2^mev). Using this approach, metrics closer to “1” better reflect the prognostic power of the model. Of note, when matched to models using either hs-TnI or sST-2, the predictive value was further enhanced by combining both biomarkers for risk prediction of cardiovascular events (CI: hs-TnI + age = 0.778 and for ST2 + age = 0.729 vs. hs-TnI + ST2 + age =0.812). Consequently, the strongest predictive power was achieved by combining the two-biomarker model with the factor age, indicating this approach as the most accurate for identification of hospitalized COVID-19 survivors at increased risk for later cardiovascular events.

## 4. Discussion

COVID-19 still represents a major burden for healthcare systems around the globe. Besides the acute phase of disease, long-term sequelae of COVID-19 pose a major challenge. While this syndrome also enhances cardiovascular risk during long-term follow-up, better pathophysiological understanding of ongoing processes with consequent identification of reliable diagnostic and prognostic tools is needed to distinguish patients at risk.

Hs-TnI and sST-2 are standard diagnostic tools used to evaluate a spectrum of cardiovascular pathologies [28,29,30,31]. Furthermore, their predictive values have been previously and extensively studied in the general population (potential control group) [32,33,34,35]. While several clinical scores and parameters have been developed to categorize COVID-19 patients according to their disease severity [36,37,38], prognostic markers for MACE in the post-COVID-19 phase are not yet available.

Therefore, the aim of the present study was to investigate novel biomarker-based risk assessment strategies for MACE during long-term follow-up. Consequently, inspired by studies which have evaluated hs-TnI and sST-2 for prediction of major cardiovascular events in patients suffering from various non-COVID-19 pathologies [28,29,31,39], sST-2 and hs-TnI levels in a single cohort of hospitalized patients with COVID-19 pneumonia were evaluated as potential prognostic markers for these events in hospitalized COVID-19 patients.

During a longer follow-up period of median 415 days, MACE was a frequent finding in our cohort. This event occurred in 8.3% of patients, with 6.7% requiring hospitalization due to cardiovascular pathology and 2.4% of patients experiencing cardiovascular death (Table 2), indicating a major cardiovascular healthcare burden in this population. However, our data are in line with recent results, which retrospectively estimated the cardiovascular risk of COVID-19 patients and mortality at 1-year-FU in comparison with contemporary and historic controls [4]. The Xie et al., authors identified an increased risk of cerebrovascular disorders, dysrhythmias, inflammatory heart disease, ischemic heart disease, heart failure, and thromboembolic disease in COVID-19 patients compared to controls. In line with the results of the present study, heart failure was the MACE with the highest incidence during long-term follow-up [4].

Interestingly, in our study, the biomarkers sST-2 and hs-TnI, besides age, were reliable predictors of MACE and significantly correlated with the time until MACE in both univariate and in the multivariate analyses (Appendix A). Of note, levels of both biomarkers at hospital admission showed a high HR (hs-TnI: HR = 4.940, 95% CI = 1.904–12.816, *p* = 0.001 and sST-2: HR = 10.901, 95% CI = 4.509–29.271, *p* < 0.001), indicating a strong correlation with cardiovascular events during long-term follow-up (Table 4). While both markers demonstrated reliable results for cardiovascular risk prediction, the best predictive power was achieved using a multi-marker model which combined hs-TnI and sST-2. This is in line with previous studies exploring biomarker risk prediction strategies in diverse cardiovascular pathologies [18,40]. While for other cardiovascular diseases, implementation of biomarker strategies led to improvement in medical care and cost-effectiveness [41], similar effects might be speculated for the investigated approach in COVID-19 patients. Indeed, identification of COVID-19 survivors at risk might promote improvement of preventive strategies with potential reduction in cardiovascular morbidity and reduction of healthcare costs. Nevertheless, while the application of preventive strategies was not investigated in our population, this speculation should be regarded with caution.

The observed correlation between hs-TnI and MACE during follow-up is in agreement with previous trials that associated higher hs-TnI concentrations with subsequent cardiovascular endpoints in cardiovascular non-COVID-19 populations [23,24,25]. In the context of the COVID-19 pandemic, elevated troponin levels as an indicator for myocardial injury were shown to correlate with 30-day mortality [26]. Nevertheless, to the best of our knowledge, our data are the first to associate increased hs-TnI with post-COVID-19 long-term cardiovascular outcomes.

Besides hs-TnI, we also identified sST-2 as a potential novel biomarker for prediction of MACE in COVID-19 patients. This is in line with previous studies in non-COVID-19 patients, which confirmed sST-2 as a reliable MACE marker [27] and found sST-2 to be superior when compared to hs-TnI as a predictor for MACE [28]. However, only a few studies have assessed sST-2 as a marker for disease severity in COVID-19 patients. Omland et al. observed elevated sST-2 levels in 74% of hospitalized COVID-19 patients in a small cohort study and found this marker to correlate with other cardiovascular biomarkers, the necessity of mechanical ventilation, and in-hospital mortality [29]. In contrast to the present study, Omland et al. assessed patients within the first ten days after hospitalization, and hence their results reflect short-term outcomes. In another smaller observational study, Zeng et al., found sST-2 levels to be increased in COVID-19 patients with both mild and severe disease courses, and while sST-2 correlated with inflammatory parameters, no correlation analysis with cardiovascular events was conducted [30]. To the best of our knowledge, the present study is the first indicating a correlation of sST-2 levels with long-term cardiovascular outcomes of COVID-19 patients.

The pathophysiological reasons for our observations remain speculative. As a marker of cardiac injury, higher hs-TnI levels were associated with diverse cardiovascular pathologies, including heart failure decompensation, myocardial infarction, and viral myocarditis [42,43], which were all reported in COVID-19 [44], thus indicating a possible link between acute myocardial injury and long-term cardiovascular risk in COVID-19 patients. sST-2 is a member of the interleukin-1 receptor family involved in inflammation with consequent cardiovascular stress processes [17]. Consequently, higher sST-2 levels might be related to higher inflammatory burden with consequent ongoing inflammation and cardiac stress metabolism. Besides the soluble isoform, a second ST-2 variant presenting as a receptor bound to the cell membrane has been reported [45]. Cardiac involvement of both isoforms is dependent on interleukin-33. During cardiac stress processes, interleukin-33 binds to the ST-2 receptor in order to reduce cardiac damage. However, while tethered with sST-2, interleukin-33 is unable to be involved in the further cellular pathway resulting in the potential loss of its cardioprotective characteristics [46]. Similar to cardiac pathologies, the interleukin-33/ST-2 axis seems also to be crucially involved in the pathophysiology of COVID-19 disease. In COVID-19, this axis is suspected to reduce responses of antiviral interferons, induce viral inflammation, and enhance thrombosis [47]. Consequently, it may promote cardiovascular events or even long-COVID-19 syndrome by increased thrombotic burden through virus persistence. The second speculation is further consolidated by a recent report, in which expression of interleukin-33 levels is correlated with seropositivity in COVID-19 convalescent individuals [47]. Nevertheless, while we did not measure serum concentrations of interleukin-33 in our cohort, this speculation should be viewed with caution. On the other hand, elevated levels of sST-2 are also indicative of increased long-term risk in patients with heart failure, myocardial infarction, and stable coronary heart disease [48]. Therefore, higher levels of sST-2 in our cohort could be related to pronounced cardiovascular comorbidities, indicating a sicker patient collective. Furthermore, sST-2 is also associated with cardiac fibrosis [49]. Consequently, pathologic cardiac remodeling, including cardiac fibrosis, could promote undesirable cardiovascular long-term effects after a COVID-19 infection.

## 5. Conclusions

In conclusion, in hospitalized COVID-19 survivors, elevated levels of hs-TnI and ST-2 at admission seem to be strong predictors of MACE during long-term follow-up. A multi-marker approach might enhance the accuracy of prediction, indicating the involvement of both proteins in post-acute sequelae of COVID-19 and potentially allowing identification of hospitalized COVID-19 survivors at increased risk for cardiovascular events. Thus, biomarker-based strategies seem to be a promising tool for risk assessment of post-acute cardiovascular sequelae of COVID-19.

## 6. Limitations

This study has some limitations which need to be addressed. The comparably small sample size might limit the reliability and validity of our results. Therefore, the findings have to be considered as primarily hypothesis-generating. Furthermore, one of the major limitations was the lack of a control group. Indeed, analysis of a control group would improve the reliability and validity of our study. As the study was conducted at a single center and exclusively with hospitalized patients, information on hs-TnI and sST-2 levels in patients with a milder or severe disease course using a multicenter design would be required to generalize the utility of these markers for risk prediction in all COVID-19 patients. Furthermore, biomarker levels were only assessed at admission, while measures during follow-up are missing. Therefore, we only studied their prognostic implementation at baseline but not as tools for disease and therapy monitoring during follow-up. Our included patients suffered from COVID-19 from June to August 2020. Consequently, rapid evolution of COVID-19 management and of the viral genome should be taken into account. We did not perform cardiac imaging, which would provide more detailed information about cardiac function and its correlation with the studied biomarker levels. In addition, cardiovascular risk prevention strategies during follow-up were not investigated in our study. More detailed molecular analyses of potentially involved pathophysiological pathways would provide a better understanding of the underlying processes involved.

## Figures and Tables

**Figure 1 biomedicines-10-02889-f001:**
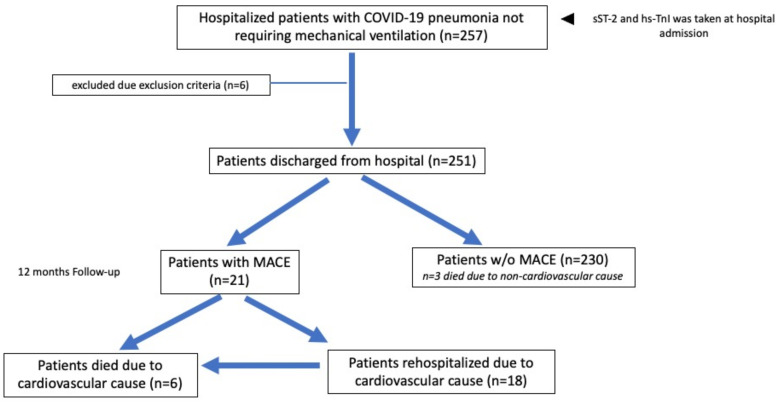
Design of the study.

**Figure 2 biomedicines-10-02889-f002:**
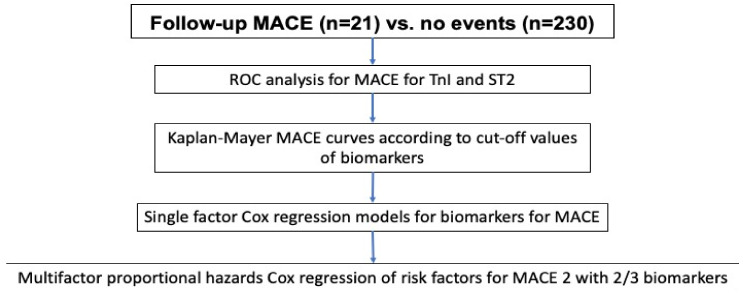
Mathematical model of the statistical analyses. ROC—receiver operator characteristics, sST-2—soluble suppression of tumorigenicity 2, hs-TnI—high sensitive troponin I.

**Figure 3 biomedicines-10-02889-f003:**
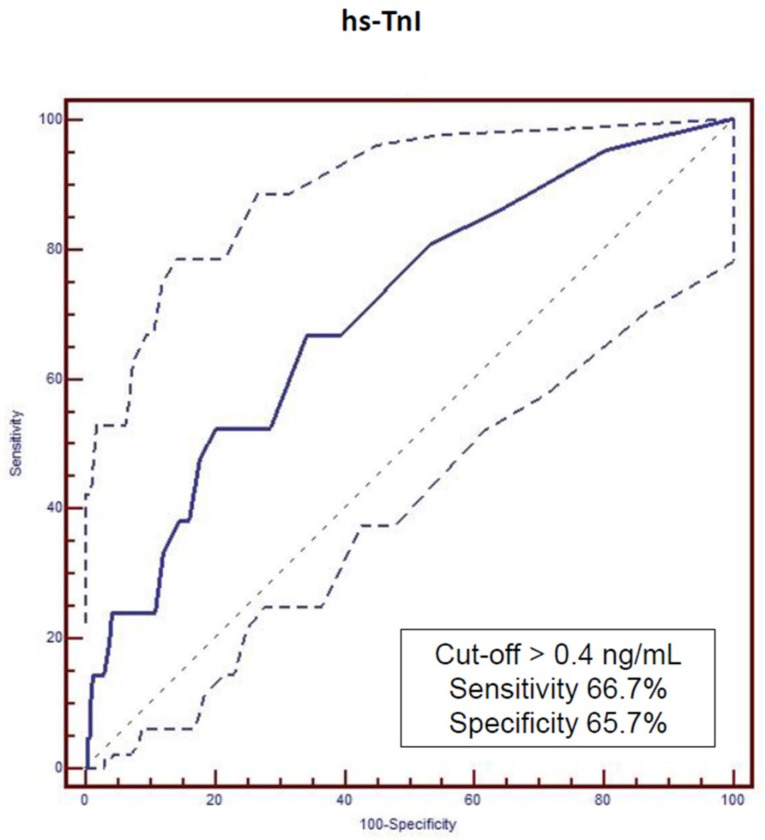
Major adverse cardiovascular events (MACE) cut-off values of high sensitive troponin I (hs-TnI) estimated by ROC analysis.

**Figure 4 biomedicines-10-02889-f004:**
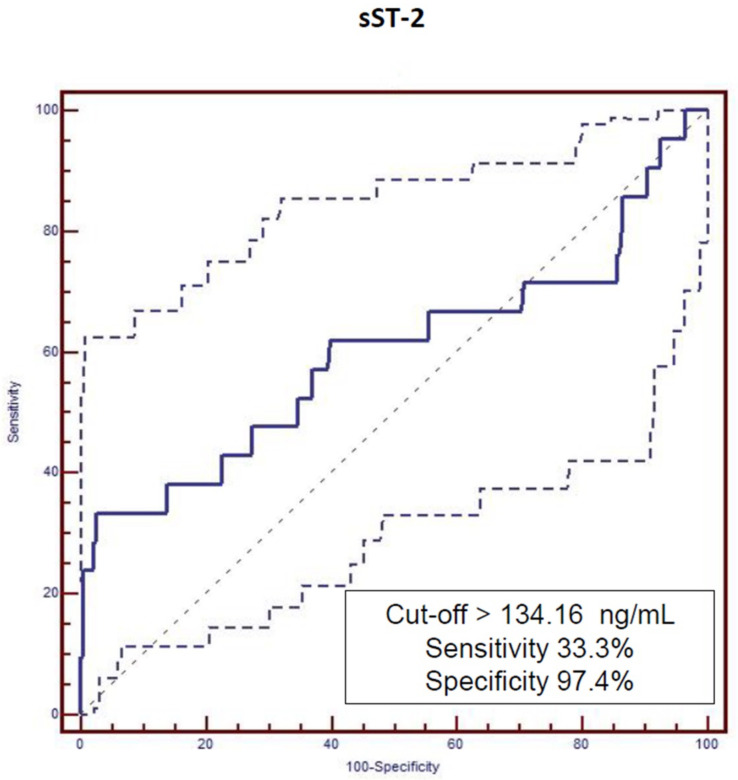
Major adverse cardiovascular events (MACE) cut-off values of soluble suppression of tumorigenicity-2 (sST-2) estimated by ROC analysis.

**Figure 5 biomedicines-10-02889-f005:**
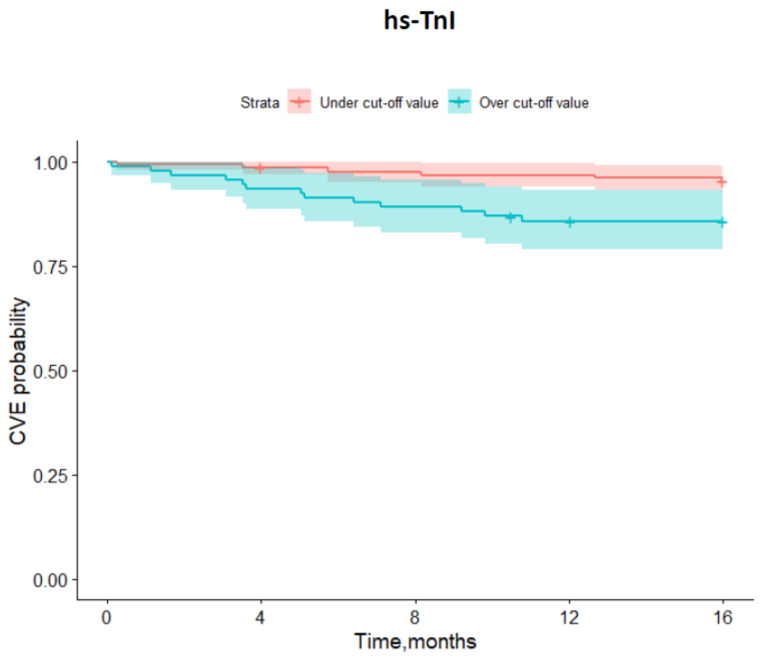
Kaplan–Meier survival curves for major adverse cardiovascular events (MACE) in patients discharged after hospitalization for COVID-19 during follow-up over one year according to under- and over cut-off values for high sensitive troponin I (hs-TnI).

**Figure 6 biomedicines-10-02889-f006:**
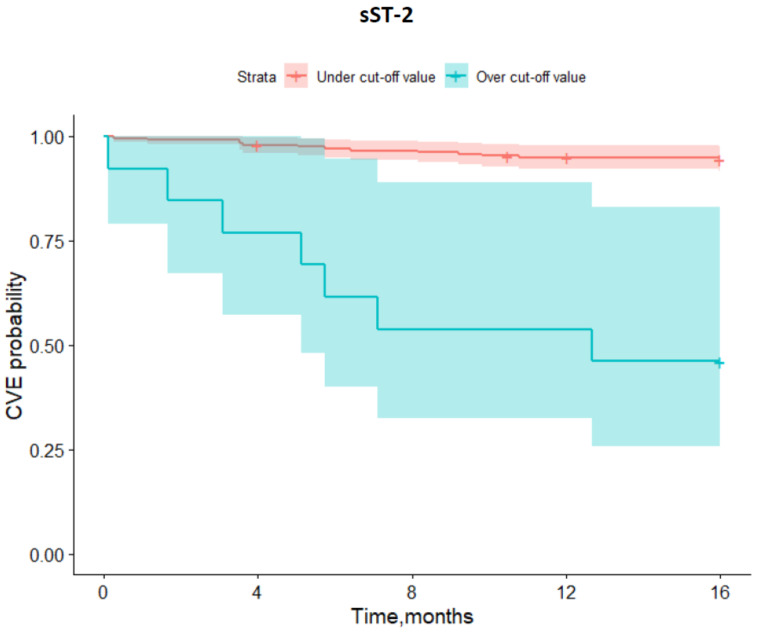
Kaplan–Meier survival curves for major adverse cardiovascular events (MACE) in patients, discharged after hospitalization for COVID-19 during follow-up over one year according to under- and over cut-off values for soluble suppression of tumorigenicity-2 (sST-2).

**Table 1 biomedicines-10-02889-t001:** Baseline characteristics.

Parameter	(Q1; Q3) or %
*n*, %	251
FU, days	415 (403; 422)
Gender, m/f	104/147 (41.4/58.6)
Age, years	59 (49; 66)
BMI, kg/m^2^	27.8 (25.0; 32.2)
Clinical presentation at admission	
SpO_2_, %	97 (95; 98.5)
Temperature at admission, °C	36.7 (36.5; 37.45)
SAP, mm Hg	130 (120; 140)
DAP, mm Hg	83 (79; 90)
BR, breaths/min	19 (18; 20)
Lung tissue damage on CT, %	36 (22; 52)
Relevant concomitant disease	
AH, % (*n*)	37.8 (95)
DM, % (*n*)	6.4 (16)
CKD, % (*n*)	1.2 (3)
CHD, % (*n*)	4.4 (11)
CHF, % (*n*)	2.0 (5)
History of MI, % (*n*)	0
History of Stroke, % (*n*)	0
Obstructive lung disease, % (*n*)	8.4 (21)
AF, % (*n*)	0.8 (2)
Laboratory parameters	
Hb, dg/L	12.8 (11.9; 12.8)
WBC, * 10^9^	4.48 (3.6; 6.1)
Platelets, * 10^9^	224 (172.3; 276.5)
CRP, mmol/L	37.9 (18; 75.9)
Procalcitonin, ng/mL	0.09 (0.05; 0.15)
Albumin, g/L	40.5 (38.2; 42.5)
CK, *n* (%)	118 (72; 215.8)
GFR, ml/min/m^2^	66.5 (57.4; 78.1)
Sodium, mmol/L	144 (141; 145)
Potassium, mmol/L	4.2 (3.9; 4.5)
Serum cardiovascular biomarkers	
sST-2, ng/mL	51.6 (31.9; 76.4)
hs-TnI, ng/mL	0.03 (0.01; 0.07)

AH—arterial hypertension, CK—creatine kinase, CHD—coronary heart disease, CHF—congestive heart failure, CKD—chronic kidney disease, CRP—C-reactive protein, CT—computer tomography, CV—cardiovascular, DAP—diastolic arterial blood pressure, DM—Diabetes Mellitus type 2, Hb—hemoglobin, SAP—systolic arterial blood pressure, WBC—white blood count, CK—creatine kinase, sST-2—solutable suppression of tumorigenicity 2, hs-TnI—high sensitive Troponin I, *—multiplication.

**Table 2 biomedicines-10-02889-t002:** Characteristics of MACE during follow-up.

Study Endpoint	% (*n*)
MACE % (*n*)	8.3 (21)
Cardiovascular hospitalization	6.7 (18)
Hospitalization for MI/ACS	0.4 (1)
Hospitalization for HF	4.8 (12)
Hospitalization for arrhythmia	0 (0)
Hospitalization for PE	0.4 (1)
Hospitalization for stroke	1.2 (3)
All cardiovascular death % (*n*)	2.4 (6)
SCD	1.2 (3)
Death due to HF	0.4 (1)
Death due to stroke	0.8 (2)

MACE—major adverse cardiovascular event, ACS—acute coronary syndrome, HF—heart failure, MI—myocardial infarction, PE—pulmonary embolism, SCD—Sudden cardiac death.

**Table 3 biomedicines-10-02889-t003:** Comparison of MACE vs. no MACE during follow-up after hospital discharge in COVID-19 survivors.

Parameter	MACE during FU,	No MACE during FU	*p*
	(Q_1_; Q_3_) or %	(Q_1_; Q_3_) or %	
*n*, %	21	230	
Gender, m/f	12/9 (57.1/42.9)	92/138 (40/60)	1
Age, years	65 (54; 73)	59 (49; 66)	0.037
BMI, kg/m^2^	28.8 (26.0; 31.6)	27.8 (25.0; 32.6)	0.726
Clinical presentation at admission			
SpO_2_, %	97 (97; 99)	97 (95; 98)	0.223
Temperature at admission, °C	36.7 (36.4; 37.6)	36.7 (36.5; 37.4)	0.946
SAP, mm Hg	140 (130; 150)	128 (120; 140)	0.014
DAP, mm Hg	88 (83; 90)	82 (79; 90)	0.033
BR, breaths/min	19 (18; 20)	19 (18; 20)	0.973
Lung tissue damage on CT, %	28 (24; 48)	32 (25; 42)	0.366
Relevant concomitant disease			
AH, % (*n*)	52.4 (11)	33.5 (84)	0.152
DM, % (*n*)	4.7 (1)	6.0 (15)	0.743
CKD, % (*n*)	0	1.2 (3)	1
CHD, % (*n*)	0	4.4 (11)	1
CHF, % (*n*)	0	2.0 (5)	1
History of MI, % (*n*)	0	0	-
History of Stroke, % (*n*)	0	0	-
Obstructive lung disease, % (*n*)	4,7 (1)	8.0 (20)	0.504
AF, % (*n*)	4,7 (1)	0.4 (1)	0.122
Laboratory parameters			
Hb, dg/L	13.3 (12.3; 13.9)	12.8 (11.9; 13.8)	0.231
WBC, * 10^9^	4.48 (3.6; 5.26)	4.48 (3.55; 6.21)	0.666
Platelets, * 10^9^	246 (175; 293)	221 (172; 275)	0.383
CRP, mmol/L	33 (19; 48)	26 (18; 40)	0.547
Procalcitonin, ng/mL	0.09 (0.06; 0.16)	0.08 (0.05; 0.15)	0.583
Albumin, g/L	40.4 (38.6; 43.8)	40.5 (38.2; 42.5)	0.591
CK, *n* (%)	155 (93; 234)	112 (70; 213)	0.999
GFR, mL/min/m^2^	68.4 (57.9; 84.0)	66.3 (57.4; 78.1)	0.382
Sodium, mmol/L	142 (141; 145)	144 (141; 145)	0.412
Potassium, mmol/L	4.2 (3.9; 4.4)	4.2 (3.9; 4.5)	0.807
Serum cardiovascular biomarkers			
sST-2, ng/mL	62.3 (27.0; 174.3)	50.3 (32.1; 72.4)	**0.141**
hs-TnI, ng/mL	0.08 (0.03; 0.12)	0.03 (0.01; 0.07)	**0.021**

AH—arterial hypertension, CK—creatine kinase, CHD—coronary heart disease, CHF—congestive heart failure, CKD—chronic kidney disease, CRP—C-reactive protein, CT—computer tomography, CV—cardiovascular, DAP—diastolic arterial blood pressure, DM—Diabetes Mellitus type 2, FU—follow-up, Hb—hemoglobin, MACE—major adverse cardiovascular event, SAP—systolic arterial blood pressure, WBC—white blood count, sST-2—solutable suppression of tumorigenicity 2, hs-TnI—high sensitive Troponin I; bold print = *p* < 0.05; *—multiplication.

**Table 4 biomedicines-10-02889-t004:** Multimarker model for prediction of MACE during follow-up after hospital discharge in COVID-19 survivors.

Risk Factors	Coef ± SE	HR	95% CI	*p*
Age + Gender + sST-2 model				
Age	0.038 ± 0.018	1.038	1.003–1.076	**0.036 ***
Male Gender	0.310 ± 0.476	1.363	0.536–3.462	0.515
sST-2	2.335 ± 0.505	10.325	3.836–27.790	**<0.001 *****
CI = 0.729, R^2^_mev_ = 0.61, R^2^_mer_ = 0.72				
Age + Gender + hs-TnI model				
Age	0.055 ± 0.018	1.056	1.020–1.094	**0.002 ****
Male Gender	0.900 ± 0.456	2.459	1.006–6.010	**0.049 ***
Hs-TnI	1.525 ± 0.477	4.593	1.803–11.700	**0.001 ****
CI = 0.778, R^2^_mev_ = 0.51, R^2^_mer_ = 0.63				
Age + Gender + sST-2 + hs-TnI model				
Age	0.046 ± 0.017	1.047	1.012–1.084	**0.009 ****
Male Gender	0.642 ± 0.480	1.900	0.742–4.870	0.181
sST-2	2.389 ± 0.504	10.901	4.059–29.271	**<0.001 *****
Hs-TnI	1.597 ± 0.486	4.940	1.904–12.816	**0.001 ****
Quality metrics: CI = 0.812, R^2^_mev_ = 0.77, R^2^_mer_ = 0.84				

CI = confidence interval, Coef = coefficient, HR = hazard ratio, MACE—major adverse cardiovascular events, SE = standard error; bold print = *p* < 0.05, *, **, ***—significance in *p* < 0.05, *p* < 0.01, *p* < 0.001, correspondingly.

## Data Availability

Naufal Zagidullin, Department of Internal Diseases, Bashkir State Medical University, Lenin str., 3, 450008, Ufa, Russia; email: znaufal@mail.ru.

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
