# Peer review of "Investigation of hs-TnI and sST-2 as Potential Predictors of Long-Term Cardiovascular Risk in Patients with Survived Hospitalization for COVID-19 Pneumonia"

_biomedicines, 2022, doi:10.3390/biomedicines10112889_

Round 1
Reviewer 1 Report
The manuscript entitled “Hs-TnI and sST-2 predict long-term cardiovascular risk in discharged hospitalized COVID-19 survivors” presents the effect of selected biomarkers on MACE in the long-term follow-up of COVID-19 patients.
The study is important because of the large number of patients who have had COVID-9.
The topic is as of interested to the area, here are my comments:
1. in Table 1 and 3 - there is the abbreviation ESR, this parameter is not in the results. Similar in the table 3 is in unused abbreviation BA.
2. I do not understand what the Hb result in Table 1 12.8 (119-128) does (???). Similar in Table 3 in the “No MACE during FU” column.
3. In my version of the manuscript, the order of the sections has changed. First, "1. Introduction", then "2. Results", then "3. Discussion" and "4. Materials and methods: 2.1 Study design”.
Author Response
Comment 1: The manuscript entitled “Hs-TnI and sST-2 predict long-term cardiovascular risk in discharged hospitalized COVID-19 survivors” presents the effect of selected biomarkers on MACE in the long-term follow-up of COVID-19 patients.
The study is important because of the large number of patients who have had COVID-9.
Response 1: We thank the reviewer for the positive evaluation of our work!
Comment 2: The topic is as of interested to the area, here are my comments:
Response 2: Thank you very much for the assessment of our work!
Comment 3: In Table 1 and 3 - there is the abbreviation ESR, this parameter is not in the results. Similar in the table 3 is in unused abbreviation BA.
Response 3: We apologize for these mistakes! We deleted the unused abbreviation from Table 1 and 3.
Comment 4. I do not understand what the Hb result in Table 1 12.8 (119-128) does (???). Similar in Table 3 in the “No MACE during FU” column.
Response 4: We also apologize for this error and have now corrected the issue. We reviewed our data and corrected the missing comma sign.
Comment 5. In my version of the manuscript, the order of the sections has changed. First, "1. Introduction", then "2. Results", then "3. Discussion" and "4. Materials and methods: 2.1 Study design”.
Response 5: We are sorry for this mix-up which must have occurred during the submission. This issue was corrected.

Reviewer 2 Report
The research targets the COVID-19 survivors and investigates a couple of markers that are useful for cardiovascular risk assessment. However, these markers don’t seem to distinguish themselves in any way in monitoring COVID-19 patients versus other categories of patients, as the title suggests and it is trying to imply. There are no data to show that these markers prove superior in monitoring COVID-19 patients vs other non-COVID-19 pathologies. Hence, in my opinion, a comparison with non-COVID-19 patients is mandatory in order to justify the claim you make within the title.
Other than that, the data are clear and the statistical analysis is sound.
Author Response
Comment 1: The research targets the COVID-19 survivors and investigates a couple of markers that are useful for cardiovascular risk assessment. However, these markers don’t seem to distinguish themselves in any way in monitoring COVID-19 patients versus other categories of patients, as the title suggests and it is trying to imply. There are no data to show that these markers prove superior in monitoring COVID-19 patients vs other non-COVID-19 pathologies. Hence, in my opinion, a comparison with non-COVID-19 patients is mandatory in order to justify the claim you make within the title.
Response 1: Thank you for bringing up this important issue for discussion! Both biomarkers hs-TnI and sST-2 are standard diagnostic tools used to evaluate a spectrum of cardiovascular pathologies[1–4]. Furthermore, their predictive values have been previously and extensively studied in the general population (potential control group) [5–8](zitate). Thus, the selection of our study design was based on well-cited studies which have evaluated hs-TnI and sST-2 for prediction of major cardiovascular events in patients suffering from various non-COVID-19 pathologies (zitate)[1,3,4,9] Inspired by these trials, after evaluating biomarker levels at hospital admission, we investigated a single cohort in follow-up using multiple regression analysis in the absence of a control group. Therefore, our intention was not to evaluate for superiority between the investigated biomarkers in COVID-19 versus non-COVID-19 patients, but rather to test the hypothesis if these biomarkers could also be applied for prediction of long-term cardiovascular risk in patients hospitalized for COVID-19. To note, to the best of our knowledge, this question has not yet been investigated. We apologize for not clarifying this issue sufficiently in our title. Consequently, to avoid any misleading of the reader, we have revised the title in the new version of our manuscript “Investigation of hs-TnI and sST-2 as potential predictors of long-term cardiovascular risk in patients with survived hospitalization for COVID-19 pneumonia.”´ Furthermore, our abstract, the introduction, the methods and the discussion section has also been revised to elucidate this issue. We hope the reviewer will be satisfied with this approach.
Nevertheless, we agree with the reviewer´s suggestion that analysis of a control group would improve the reliability and validity of our study. However, as already mentioned in the limitations section of our manuscript, our results were evaluated during the “first wave” of the pandemic. This period was characterized by demands on medical system resources, which consequently affected cardiovascular outcomes in the general population. Thus, as already implemented in previous well citied trials evaluating cardiovascular risk during post COVID-19 sequelae, under optimal circumstances, evaluation of long-term cardiovascular outcomes in COVID-19 survivors should include follow-up of non-COVID-19 control group from the same time period as in the investigated COVID-19 cohort[10]. Unfortunately, during the “first wave” of the pandemic, we did not draw samples for biomarker analyses from non-COVID-19 patients, and therefore, we are unable to provide a reliable control cohort. Indeed, this is a major limitation of our study, which has now been discussed in the limitations section of our manuscript. We hope the reviewer will be satisfied with this approach. Again, we thank the reviewer for raising this important issue, which inspired us to incorporate the suggested study design in future studies assessing the use of cardiovascular biomarker analyses in COVID-19 patients.
Literature
- Omland, T.; Pfeffer, M.A.; Solomon, S.D.; De Lemos, J.A.; Røsjø, H.; Benth, J.Š.; Maggioni, A.; Domanski, M.J.; Rouleau, J.L.; Sabatine, M.S.; et al. Prognostic value of cardiac troponin I measured with a highly sensitive assay in patients with stable coronary artery disease. J. Am. Coll. Cardiol. 2013, 61, 1240–1249, doi:10.1016/J.JACC.2012.12.026.
- Richards, A.M. ST2 and Prognosis in Chronic Heart Failure. J. Am. Coll. Cardiol. 2018, 72, 2321–2323, doi:10.1016/J.JACC.2018.08.2164.
- Gaggin, H.K.; Szymonifka, J.; Bhardwaj, A.; Belcher, A.; De Berardinis, B.; Motiwala, S.; Wang, T.J.; Januzzi, J.L. Head-to-head comparison of serial soluble ST2, growth differentiation factor-15, and highly-sensitive troponin T measurements in patients with chronic heart failure. JACC. Heart Fail. 2014, 2, 65–72, doi:10.1016/J.JCHF.2013.10.005.
- Emdin, M.; Aimo, A.; Vergaro, G.; Bayes-Genis, A.; Lupón, J.; Latini, R.; Meessen, J.; Anand, I.S.; Cohn, J.N.; Gravning, J.; et al. sST2 Predicts Outcome in Chronic Heart Failure Beyond NT-proBNP and High-Sensitivity Troponin T. J. Am. Coll. Cardiol. 2018, 72, 2309–2320, doi:10.1016/J.JACC.2018.08.2165.
- Ho, J.E.; Sritara, P.; Defilippi, C.R.; Wang, T.J. Soluble ST2 testing in the general population. Am. J. Cardiol. 2015, 115, 22B-25B, doi:10.1016/J.AMJCARD.2015.01.036.
- Zhang, T.; Xu, C.; Zhao, R.; Cao, Z. Diagnostic Value of sST2 in Cardiovascular Diseases: A Systematic Review and Meta-Analysis. Front. Cardiovasc. Med. 2021, 8, doi:10.3389/FCVM.2021.697837.
- Blankenberg, S.; Salomaa, V.; Makarova, N.; Ojeda, F.; Wild, P.; Lackner, K.J.; Jørgensen, T.; Thorand, B.; Peters, A.; Nauck, M.; et al. Troponin I and cardiovascular risk prediction in the general population: the BiomarCaRE consortium. Eur. Heart J. 2016, 37, 2428–2437, doi:10.1093/EURHEARTJ/EHW172.
- Farmakis, D.; Mueller, C.; Apple, F.S. High-sensitivity cardiac troponin assays for cardiovascular risk stratification in the general population. Eur. Heart J. 2020, 41, 4050–4056, doi:10.1093/EURHEARTJ/EHAA083.
- Rywik, T.M.; Janas, J.; Klisiewicz, A.; Leszek, P.; SobieszczaÅ„ska-MaÅ‚ek, M.; Kurjata, P.; Rozentryt, P.; Korewicki, J.; Jerzak-WodzyÅ„ska, G.; ZieliÅ„ski, T. Prognostic value of novel biomarkers compared with detailed biochemical evaluation in patients with heart failure. Pol. Arch. Med. Wewn. 2015, 125, 434–442, doi:10.20452/PAMW.2884.
- Xie, Y.; Xu, E.; Bowe, B.; Al-Aly, Z. Long-term cardiovascular outcomes of COVID-19. Nat. Med. 2022, 28, doi:10.1038/s41591-022-01689-3.

Round 2
Reviewer 2 Report
Thank you for your detailed answer. It would have been rather surprising to see that such markers, with a proven value as predictors of major cardiac risk, are not equally useful for evaluating COVID-19 patients. It is absolutely appropriate to relate to various other studies investigating hs-TnI and sST-2 and to compare data in order to stress that the results presented here are sound.